# Kernel RNN Learning (KeRNL)

**Christopher Roth**[1],[2]**, Ingmar Kanitscheider**[2],[3]**, and Ila Fiete**[2],[4]

[1] Department of Physics, University of Texas at Austin, Austin, TX, 78712
[2] Department of Neuroscience, University of Texas at Austin, Austin, TX, 78712
[3] OpenAI, San Francisco CA, 94110
[4] Department of Brain and Cognitive Sciences, Massachusetts Institute of Technology, Cambridge, MA, 02139

## Abstract

We describe Kernel RNN Learning (KeRNL), a reduced-rank, temporal eligibility trace-based approximation to backpropagation through time (BPTT) for training recurrent neural networks (RNNs) that gives competitive performance to BPTT on long time-dependence tasks. The approximation replaces a rank-4 gradient learning tensor, which describes how past hidden unit activations affect the current state, by a simple reduced-rank product of a sensitivity weight and a temporal eligibility trace. In this structured approximation motivated by node perturbation, the sensitivity weights and eligibility kernel time scales are themselves learned by applying perturbations. The rule represents another step toward biologically plausible or neurally inspired ML, with lower complexity in terms of relaxed architectural requirements (no symmetric return weights), a smaller memory demand (no unfolding and storage of states over time), and a shorter feedback time. Finally, we show that KeRNL can learn long time-scales more efficiently than BPTT in an online setting.

## 1 Introduction

Animals and humans excel at learning tasks that involve long-term temporal dependencies. A key challenge of learning such tasks is the problem of spatiotemporal credit assignment: the learner must find which of many past neural states is causally connected to the currently observed error, then allocate credit across neurons in the brain. When the time-dependencies between network states and errors are long, learning becomes difficult.

In machine learning, the current standard for training recurrent architectures is *Backpropagation Through Time* (BPTT, Rumelhart et al. (1985), Werbos (1990)). BPTT assigns temporal credit or blame by unfolding a recurrent neural network in time up to a horizon length $T$, processing the input in a forward pass, and then backpropagating the error back in time in a backward pass (see Fig. 1a).

From a biological perspective, BPTT – like backpropagation in feedforward neural networks – is implausible for many reasons. For each weight update, BPTT requires using the transpose of the recurrent weights to transmit errors backwards in time and assign credit for how past activity affected present performance. Running the network with transposed weights requires that the network either has two-way synapses, or uses a symmetric copy of the feedforward weights to backpropagate error. In either case, the network must alternatingly gate its dynamical process to run forward or backward, and switch from nonlinear to linear dynamics, depending on whether activity or errors are being sent through the network.

From both biological and engineering perspectives, there is a heavy memory demand: the complete network states, going $T$ timesteps back in time, must be stored. The time-complexity of computation of the gradient in BPTT scales like $T$, making each iteration slow when training tasks with long time scale dependencies. Although $T$ should match the length of the task or the maximum temporal lag between network states and errors for unbiased gradient learning, in practice $T$ is often truncated to mitigate these computational costs, introducing a bias.

The present work is another step in the direction of providing heuristics and relaxed approximations to backpropagation-based gradient learning for recurrent networks. KeRNL confronts the problems of efficiency and biological plausibility. It replaces the lengthy, linearized backward-flowing

backpropagation phase with a product of a forward-flowing temporal eligibility trace and a spatial sensitivity weight. Instead of storing all details of past states in memory, synapses integrate their past activity during the forward pass (see Fig. 1b). The network does not have to recompute the entire gradient at each update, as the time-scales of the eligibility trace and the sensitivity weight are learned over time.

## 2 RELATED WORK

In recent years, much work has been devoted to implementing backpropagation algorithms in a more biologically plausible way, partly in the hope that more plausible implementations might also be simpler. The symmetry requirement between the forwards and backwards weights can be alleviated by using random return weights (Lillicrap et al. (2016) and Nøkland (2016)), however, learning still requires a separate backward pass through a network with linearized dynamics. Neurons may be able to extract error information in the time derivative of their firing rates using an STDP-like learning rule (Bengio et al. (2015)), with error backpropagation computed as a relaxation to equilibrium (Scellier & Bengio (2017)), at least for learning fixed points.

Other work has focused on replacing batch learning with online learning. Typically, BPTT is implemented in a setting where data is prepared into batches of fixed sequence length $T$ and used to perform learning in a $T$-step unrolled graph; however, online learning, with a constant stream of data error signals, is a more natural description of how the world supplies a learning system with data. BPTT without truncation struggles with online learning, as it must repeatedly backpropagate the error all the way through a continuously expanding graph. Since computation of the unbiased gradient scales with the length of the graph, gradient computation increases linearly with time. For a task with $T$ timesteps, the total computation of the gradients scales like $T^2$.

Real Time Recurrent Learning (RTRL, Williams & Zipser (1989)) and Unbiased Online Gradient Optimization (UORO, Tallec & Ollivier (2017), Ollivier et al. (2015)) deal with this issue by keeping track of how the synaptic weights affect the hidden state in a feedforward way. Decoupled Neural Interfaces (DNI Jaderberg et al. (2016)) estimates the truncated part of the gradient by continually predicting the future loss with respect to the hidden state.

KeRNL offers this same advantage, in addition to other benefits. RTRL requires that the network keep track of an unwieldy rank-3 tensor, which could not be stored by any known biological entities. UORO factorizes this into rank-2 objects but still requires non-local computations like vector norm operations. Finally, DNI requires an entire separate network to keep track of the synthetic gradient. KeRNL is distinguished by its simplicity, requiring only rank-2 tensors. All computations are local, and synapses need to integrate over only a few relevant quantities.

## 3 THE LEARNING RULE

Consider a single-layer RNN in discrete time (indexed by $t$) with readout, input, and hidden layer activations given by $\mathbf{y}^t, \mathbf{x}^t$, and $\mathbf{h}^t$, respectively (boldface represents vectors, with vector entries denoting the activity of individual units). The dynamics of the recurrently connected hidden units are given by:

$$\mathbf{h}^{t+1} = \sigma\left(\mathbf{g}^{t+1}\right) = \sigma\left(W^{rec}\mathbf{h}^t + W^{in}\mathbf{x}^t + \mathbf{b}\right) \tag{1}$$

where $W^{rec}$, $W^{in}$ are the recurrent and input weights, $\mathbf{b}$ are the hidden biases, $\sigma$ is a general pointwise non-linearity, and $\mathbf{g}^t$ represents the summed inputs (pre-nonlinearity) to the neurons at time t. The readout is given by $\mathbf{y}^t = \sigma^{out}\left(W^{out}\mathbf{h}^t + \mathbf{b}^{out}\right)$. The objective function is $C = C(\{(\mathbf{x}^0, \mathbf{y}^0), \cdots (\mathbf{x}^T, \mathbf{y}^T), \hat{\mathbf{y}}^T\})$, where $\hat{\mathbf{y}}^T$ is the target output, in the case where error feedback is received at the end of an episode of length $T$, and $C = \sum_{t=0}^{T} C^t$ when errors $C^t = C(\{(\mathbf{x}^0, \mathbf{y}^0), \cdots (\mathbf{x}^t, \mathbf{y}^t), \hat{\mathbf{y}}^t\})$ are received continuously over the episode. The parameters $W^{in}, W^{rec}, W^{out}, \mathbf{b}, \mathbf{b}^{out}$ are trainable.

Assuming the readout weights and biases $(W^{out}, \mathbf{b}^{out})$ are trained in the usual way, the KeRNL update rule for the RNN weights $W = \{W^{in}, W^{rec}\}$ is simply:

$$(\Delta W_{jk})^t = -\eta \sum_i \delta_i^t \beta_{ij} e_{jk}^t, \tag{2}$$

where $\delta_i^t \equiv \frac{dC^t}{dh_i^t} = \sum_l \frac{dC^t}{dy_l^t} \frac{dy_l^t}{dh_i^t}$ is the gradient of the cost with respect to the current hidden state, $\beta_{ij}$ is a set of learned *sensitivity weights*, and

$$e_{jk}^t \equiv \sum_\tau K(\tau, \gamma_j) \sigma'(u) \Big|_{g_j^{t-\tau}} s_k^{t-\tau} \tag{3}$$

is a *local eligibility trace* (Williams (1992)) consisting of a temporally filtered version of the product of presynaptic activation and a postsynaptic activity factor. The temporal filter or kernel, $K$, in the eligibility has learnable time-scales; in this manuscript we use the simplest version of a low-pass temporal filter, a decaying exponential with a single time-constant $\gamma_j$ per neuron: $K(\tau, \gamma_j) = \exp(-\gamma_j \tau)$, though one can imagine many other function choices with multiple timescales. The role of the eligibility is to specify how strongly a synapse $W_{jk}$ should be held responsible for any errors in neuron $j$ at the present time, on the basis on how far in the past the presynaptic neuron $k$ was active. Here $s_k^{t-\tau} = \{x_k^{t-\tau}, h_k^{t-\tau-1}\}$ stands in for the activation of the neuron presynaptic to the synapse being updated. [1]. Since the eligibility trace can be computed during the forward pass, KeRNL does not require backpropagating the error through time. Furthermore, KeRNL only uses at most rank-2 tensors, so neurons and synapses could plausibly do all of the required computation. The contrast between BPTT and KeRNL is depicted in Fig. 1a,b.

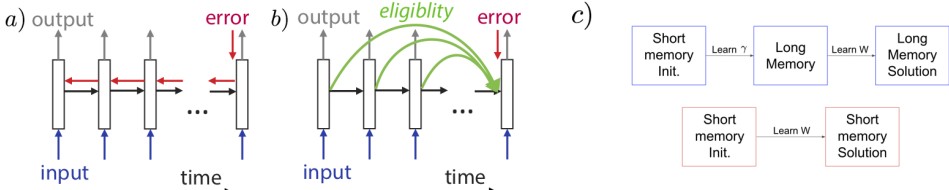

**Figure 1:** a), b) Schematic depicting the difference in the flow of information between a) BPTT, b) KeRNL. c) Schematic depicting how KeRNL (Blue) can be used to find long term memory solutions when BPTT (red) struggles

KeRNL emerges from the following Ansatz:

$$\frac{\partial h_i^t}{\partial h_j^{t-\tau}} \approx \beta_{ij} K(\tau, \gamma_j). \tag{4}$$

We call $\partial h_i^t / \partial h_j^{t-\tau}$, a key term in the computation of the gradient, the *sensitivity tensor* in an extension of the usage in Fiete & Seung (2006)). This sensitivity describes how the activity of neuron $j$ at a previous time $t - \tau$ affects the activity of neuron $i$ at the current time $t$. While the true sensitivity is a 4-index tensor summarizing many interactions based on the many paths through which activity propagates forward in a recurrent network, we approximate it with a product of a (learnable) rank-2 sensitivity weight matrix $\beta$ and a temporal kernel $K$ with (learnable) inverse-time coefficients $\gamma$. The sensitivity weights $\beta_{ij}$ describe how strongly neuron $j$ affects neuron $i$ on average, while the temporal kernel describes how far into the future the activity of a neuron affects the other neurons for learning. We describe how to learn these parameters $(\beta, \gamma)$ in the next section.

We arrive at KeRNL by using our Ansatz for the sensitivity (4) in the computation of a gradient-based weight update, instead of using the true sensitivity. First, we write down the full gradient rule for a recurrent network. If the parameters $W_{ij}$ are treated as functions that can vary over time during a trial, then the derivative can be written as a functional derivative:

$$\frac{dC}{dW_{jk}} = \frac{1}{T} \sum_{t=0}^{T} \frac{\delta C}{\delta W_{jk}(t)}. \tag{5}$$

This is simply mathematical notation for the "unfolding-in-time" trick, in which the network and weights are assumed to be replicated for each time-step of the dynamics of a recurrent network,

---

[1]The rule to update the hidden layer biases is given by $(\Delta b_i)^t = -\eta \sum_i \delta_i^t \beta_{ij} e_j^t$ where $e_j^t = \sum_\tau \exp(-\gamma_j \tau) \sigma'(g_j^{t-\tau})$ integrates over a clamped input.

and separate gradients are computed for each time-replica of the weights; the actual weight updates are simply the average of the separate weight variations for each time-replica. We next apply the sensitivity lemma Fiete & Seung (2006) to express gradients with respect to weights as gradients with respect to input activations, times the presynaptic activity: $\frac{\delta C}{\delta W_{jk}(t)} = \frac{\delta C}{\delta g_j(t)} s_k(t)$. Thus,

$$\frac{dC}{dW_{jk}} = \sum_\tau \frac{\delta C}{\delta g_j(t-\tau)} s_k(t-\tau) \quad = \quad \sum_i \sum_\tau \frac{\delta C}{\delta h_i(t)} \frac{\delta h_i(t)}{\delta g_j(t-\tau)} s_k(t-\tau) \tag{6}$$

$$= \quad \sum_i \sum_{\tau=0}^{t} \frac{\delta C}{\delta h_i(t)} \frac{\delta h_i(t)}{\delta h_j(t-\tau)} \sigma'(u)\Big|_{g_j(t-\tau)} s_k(t-\tau) \tag{7}$$

By replacing the sensitivity $\delta h_i(t)/\delta h_j(t-\tau)$ with our Ansatz (4), we arrive at our learning rule, KeRNL(2).

The time-dependent part of the computation– a leaky integral of the product of the presynaptic activity multiplied by the instantaneous change in the postsynaptic activity– can be computed during the forwards pass, without any backpropagation of activity or error signals.

## 4    LEARNING THE SENSITIVITY WEIGHTS AND INVERSE-TIMESCALES

For our Ansatz to align as well as possible with the gradient, we allow the sensitivity weights $\beta$ and inverse-timescales $\gamma$ to be learned. We learn these parameters by tracking the effect of small i.i.d. hidden perturbations $\xi$ during the forward pass. In order to do so our hidden neurons must store two values, the true hidden state $\mathbf{h}$, and a perturbed hidden state $\tilde{\mathbf{h}}$, which is generated by applying noise to the neurons during the forward pass:

$$\tilde{\mathbf{h}}^{t+1} = \sigma\left(W^{rec}(\tilde{\mathbf{h}}^t + \xi^t) + W^{in}\mathbf{x}^t + \mathbf{b}\right), \tag{8}$$

The effect of previous noise on the current hidden state can be computed using the sensitivity

$$\tilde{h}_i^t - h_i^t = \sum_\tau \sum_j \frac{\partial h_i^t}{\partial h_j^{t-\tau}} \xi_j^{t-\tau} \tag{9}$$

We train $\gamma, \beta$ to predict the network's response to these noisy perturbations. We take gradients with respect to the objective function.

$$C_\beta = \left(\tilde{h}_i^t - h_i^t - \sum_j \beta_{ij} \sum_\tau \exp(-\gamma_j \tau)\xi_j^{t-\tau}\right)^2 \tag{10}$$

which we have generated by substituting our Ansatz (4) into (9). [2]

Taking gradients with respect to this objective function gives us the following update rule for the sensitivity weights and inverse-timescales.

$$\Delta\beta_{ij}^t = -\eta_m \delta_{\beta_i}^t \Omega_j^t(\xi)$$
$$\Delta\gamma_j^t = -\eta_m \sum_i \delta_{\beta_i}^t \beta_{ij} \Gamma_j^t(\xi) \tag{11}$$

Here $\delta_{\beta_i}{}^t \equiv \sum_j \beta_{ij}\Omega_j^t - (\tilde{h}_i^t - h_i^t)$ represents the error in reconstructing the effect of the perturbation via the sensitivity weights and $\Omega_j^t(\xi) \equiv \sum_\tau \exp(-\gamma_j\tau)\xi_j^{t-\tau}$ and $\Gamma_j^t(\xi) \equiv \frac{d\Omega_j^t}{d\gamma_j} = -\sum_\tau \tau \exp(-\gamma_j\tau)\xi_j^{t-\tau}$ are integrals that neuron $h_j$ performs over the applied perturbation $\xi$. In our implementation, we update these parameters immediately before we compute the gradient using (2). The full update rule is described in the pseudocode table.

---

[2]If we don't care about the size of the gradients and only the direction, we can use the cost function $C_\beta = -\frac{(\tilde{\mathbf{h}}^t - \mathbf{h}^t) \cdot \mathbf{O}^t}{||\tilde{\mathbf{h}}^t - \mathbf{h}^t||\,||\mathbf{O}^t||}$ where $O_i^t = \sum_j \beta_{ij} \sum_\tau \exp(-\gamma_j\tau)\xi_j^{t-\tau}$. This cost function trains the parameters to predict the correct direction of the perturbed hidden state minus the hidden state and works for algorithms where the gradient is divided by a running average of its magnitude (RMSProp, Adam).

---

**Algorithm 1** Pseudocode table describing the implementation of Online-KeRNL on an RNN. For Batched-KeRNL we only update parameters when t = T

---

**while** $t \leq T$ **do**

$\quad \mathbf{h}^t \leftarrow \sigma \left( W \mathbf{h}^{t-1} + W^{in} \mathbf{x}^{t-1} + \mathbf{b} \right)$    /* propagate data forwards */

$\quad \tilde{\mathbf{h}}^t \leftarrow \sigma \left( W(\tilde{\mathbf{h}}^{t-1} + \xi^{t-1}) + W^{in} \mathbf{x}^{t-1} + \mathbf{b} \right)$    /*propagate noisy network forwards */

$\quad \Omega_j{}^t \leftarrow \exp(-\gamma_j)\Omega_j{}^{t-1} + \xi_j{}^t$    /*Integrate over perturbations*/

$\quad \Gamma_j{}^t \leftarrow \exp(-\gamma_j)\Gamma_j{}^{t-1} - \exp(-\gamma_j)\Omega_j{}^{t-1}$    /*Derivative of $\mathbf{\Omega}^t$ w.r.t. $\gamma^t$ */

$\quad e_{jk}^t \leftarrow \exp(-\gamma_j)e_{jk}^{t-1} + \sigma'(\mathbf{g}_j^t)s_k^t$    /*Update eligibility traces*/

$\quad \delta_{\beta_i}^t \leftarrow \sum_j \beta_{ij}\Omega_j^t - (\tilde{h}_j^t - h_j^t)$    /* Calculate error in predicting effect of perturbations */

$\quad \beta_{ij}^t \leftarrow \beta_{ij}^{t-1} - \eta_m \delta_{\beta_i}^t \Omega_j^t$    /*Update sensitivity weights */

$\quad \gamma_j^t \leftarrow \gamma_j^{t-1} - \eta_m \sum_i \delta_{\beta_i}^t \beta_{ij}\Gamma_j^t$    /*Update kernel coefficients */

$\quad \delta_i^t = \sum_l \frac{dC}{dy_l^t} W_{li}^{out}$    /*Compute error in hidden state */

$\quad W_{jk}^t \leftarrow W_{jk}^{t-1} - \eta \sum_i \delta_i^t \beta_{ij} e_{jk}^t$    /*Compute and apply gradients*/

**end while**

---

## 5 EMPIRICAL RESULTS

We test KeRNL on several benchmark tasks that require memory and computation over time, showing that it is competitive with BPTT across these tasks. We implemented batch learning with KeRNL and BPTT on two tasks: the adding problem ( Hochreiter & Schmidhuber (1997); Hochreiter et al. (2001)) and pixel-by-pixel MNIST (LeCun et al. (1998)). We implemented an online version of KeRNL with an LSTM network on the $A^n, B^n$ task (Gruslys et al. (2016)) to compare with results from the UORO algorithm (Tallec & Ollivier (2017)). [3]

The tuned hyperparameters for BPTT and KeRNL were the learning rate, $\eta$, and the gradient clipping parameter, gc (Pascanu et al. (2013)). For KeRNL, we additionally permitted a shared learning rate parameter for the sensitivity weights and kernels, $\eta_m$. In practice, the same hyperparameter settings $\eta$, gc tended to work well for both BPTT and KeRNL. The additional hyperparameter for KeRNL, $\eta_m$, did not need to be find tuned, and often worked well across a broad range (across several orders of magnitude, so long as it not too small but smaller than $\eta$).

We implemented both the RMSprop (Tieleman & Hinton (2012)) and Adam (Kingma & Ba (2014)) optimizers and reported the best result.

### 5.1 ADDING PROBLEM

In the adding problem, the network receives two input streams, one a sequence of random numbers in $[0, 1]$, and the second a mask vector of zeros, with two entries set randomly to one in each trial. The network's task is to sum the input from the first stream whenever there is a non-zero entry in the second. This task requires remembering sparse pieces of information over long time scales and ignoring long sequences of noise, which is difficult for RNNs when the sequences are long.

We tested the performance of two networks on a variety of sequence lengths, up to 400, using both BPTT and KeRNL, Table 2. The networks were an IRNN, which is an RNN with a ReLU non-linearity where the recurrent weight matrix is initialized to identity, and a RNN with tanh non-linearity. The implementation details are described in Appendix A.

Untruncated BPTT applied to an IRNN performed very well on this task, but less so on the RNN with tanh nonlinearity. KeRNL was somewhat unstable on the IRNN, but it outperformed BPTT with the tanh nonlinearity (Fig. 2).

We believe that KeRNL outperforms BPTT on the tanh nonlinearity because our Ansatz allows the sensitivity $\frac{d\mathbf{h}^t}{d\mathbf{h}^{t-\tau}}$ to have relatively long timescales, while the BPTT sensitivities are squashed by the

---

[3]KeRNL can be implemented more generally on any circuit dynamics with Markov architecture, including LSTMs. For LSTMs, the gradient terms are slightly more complicated in form, but still easy to compute (Appendix).

| Learning Rule, Network | Algorithm | $\eta$ | gc | $\eta_m$ |
|---|---|---|---|---|
| BPTT, tanh RNN | RMS Prop | $10^{-3}$ | 100.0 | – |
| BPTT, ReLU IRNN | RMS Prop | $10^{-4}$ | 100.0 | – |
| KeRNL, tanh RNN | RMS Prop | $10^{-3}$ | 100.0 | $10^{-5}$ |

**Table 1:** Tuned hyperparameters for the adding problem with sequence length 400.

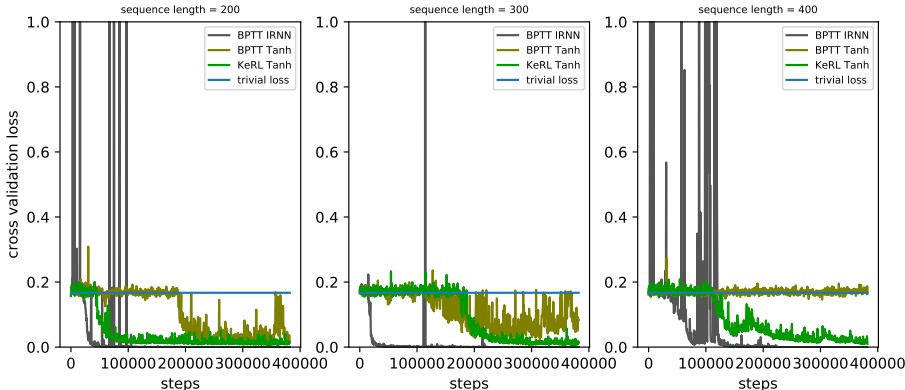

**Figure 2:** Single-trial example of cross validation loss on the adding problem for sequences of length 200,300,400.

tanh nonlinearity. By applying gradients generated by our Ansatz (instead of the true gradients) we push our network toward a solution with longer time scales via a feedback alignment-like mechanism (Lillicrap et al. (2016),Nøkland (2016)), as schematized in Fig. 1c.

To investigate the importance of learning the kernel timescales, we implemented KeRNL without training the sensitivity weights ($\beta$) or the inverse timescales ($\gamma$). When these parameters are not learned, KeRNL is still able to perform the task for the shorter 200-length sequence (Table 2) implying that a feedback-alignment-like mechanism (Lillicrap et al. (2016), Nøkland (2016)) may be enabling learning even when the error signals are not delivered along the instantaneous gradients.

For longer sequences, however, learning the sensitivity and timescale parameters is important. Surprisingly, learning the inverse timescales is even more important than learning the sensitivity weights. We hypothesize that as long as the timescales over which error is correlated with outcome are appropriate, sensitivity weights are relatively less important because of feedback-alignment-like mechanisms. We show an example of how the timescales may change in Fig. 3.

## 5.2 PIXEL-BY-PIXEL MNIST

Our second task is pixel-by-pixel MNIST (LeCun, 1998). Here the RNN is given a stream of pixels left-to-right, top-to-bottom for a given handwritten digit from the MNIST data set. At the end of the sequence, the network is tasked with identifying the digit it was shown. This problem is difficult, as the RNN must remember an long sequence of $784$ singly-presented pixels. We tuned over the same hyperparameters as in the adding problem, looking at performance after $100,000$ minibatches. Neither KeRNL nor BPTT worked well with a tanh nonlinearity, but both performed relatively well on an IRNN, Fig. 3. KeRNL preferred a slightly lower learning rate $\eta$ than BPTT.

| Learning Rule | Algorithm | $\eta$ | gc | $\eta_m$ |
|---|---|---|---|---|
| BPTT | RMSProp | $10^{-5}$ | 100.0 | - |
| KeRNL | RMSProp | $10^{-6}$ | 100.0 | $10^{-8}$ |

**Table 3:** Hyperparameters for pixel-by-pixel MNIST.

While the KeRNL algorithm is able to learn almost as quickly on pixel-by-pixel MNIST, it does not reach as high an asymptotic performance. Still, it performs reasonably well relative to BPTT on the task.

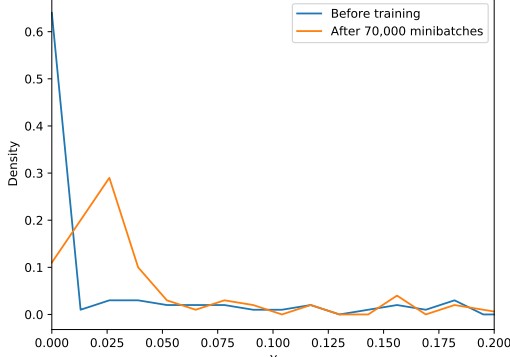

| Mode | 200 | 400 |
|---|---|---|
| Fix $\beta$, Fix $\gamma$ | 0.014 | 0.180 |
| Learn $\beta$, Fix $\gamma$ | 0.170 | 0.171 |
| Fix $\beta$, Learn $\gamma$ | 0.031 | 0.076 |
| Learn $\beta$, Learn $\gamma$ | 0.008 | 0.031 |
| BPTT | 0.020 | 0.171 |

**Figure 3 & Table 2:** Learning of KeRNL parameters. Left: Histogram of inverse time coefficients before training (blue) and after $7 \times 10^4$ minibatches (orange) on the adding problem (200): the network learns the relative importance of certain time-scales. Right: Examining the relative importance of learnable parameters in KeRNL: Performance on BPTT and various versions of KeRNL using a tanh RNN after $7 \times 10^4$ minibatches: fixing the sensitivities, $\beta$, while learning the inverse timescales, $\gamma$, is better than doing the reverse.

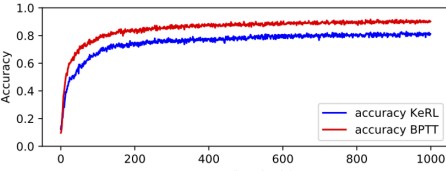 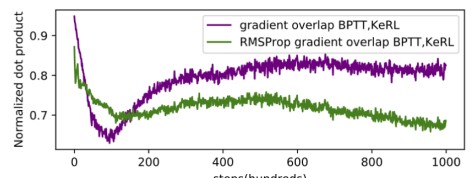

**Figure 4:** Left: Cross validation accuracy on pixel-by-pixel MNIST using BPTT (red) and KeRNL(blue). Right: Normalized dot product between gradients (purple) and RMSProp gradients (green) computed by KeRNL and BPTT

### 5.3 ONLINE KERNL

While KeRNL is comparable in speed to BPTT for batch learning, we expect it to be significantly faster for online learning when the time-dependencies are of length $T$. Untruncated BPTT requires information sent back $T$ steps in time for each weight update, thus the wallclock speed of computation of the gradients at each weight update in online learning scales as $T$, and the total scaling is thus of order $T^2$. If BPTT updates are truncated $S < T$ steps back in time, the scaling is $ST$. KeRNL requires no backward unrolling in time, thus online KeRNL requires only $O(1)$ time per weight update, for a total scaling of $T$. As a result, optimized-speed online-KeRNL should run faster than truncated online BPTT by a factor $T$ when the trunctation time is similar to the total time-dependencies in the problem.

We tested the performance of online KeRNL against UORO, another online learning algorithm, and online BPTT on the $A^n, B^n$ task, where the network must predict the next character in a stream of letters. Each stream consists first of a sequence of $n$ $A$s followed by a sequence of $n$ $B$s. The length, $n$, of the sequences is randomly generated in some range. The network cannot solve this task perfectly, as it can not predict the number of $A$s before it has seen the sequence, but can do well by matching the number of $B$s to the number of $A$s. We generated $n \in \{1, 32\}$. The minimum achievable average bit-loss for this task is 0.14.

To compare with results in the literature, we implemented KeRNL in an LSTM layer, with **h** representing a concatenation of the hidden and cell states (Details in Appendix B). Instead of optimizing common hyperparameters, we simply used the values from Tallec & Ollivier (2017), which included decaying the learning rate in time as $\eta^t = \eta/(1 + \alpha\sqrt{t})$. However, we varied the learning rate $\eta_m$, with $\eta_m^t = \eta_m/(1 + \alpha\sqrt{t})$.

Results other than those for KeRNL are from Tallec & Ollivier (2017). With very little hyperparameter tuning, online KeRNL is able to do very well on the $A^n, B^n$ task, coming close to the minimum

| Algorithm-Optimizer | $\eta$ | $\eta_m$ | $\alpha$ |
|---|---|---|---|
| KeRNL-Adam | $10^{-3}$ | $10^{-2}$ | 0.03 |

**Table 4:** $A^n, B^n$ hyperparameters

| Algorithm | KeRNL | 1 Step BPTT | 2 Steps BPTT | 16 Step BPTT | UORO |
|---|---|---|---|---|---|
| Bit Loss | 0.149 | 0.178 | 0.149 | 0.144 | 0.147 |

**Table 5:** Average cross-entropy bit-loss (over $10^4$ minibatches) on the online $A^n, B^n$ task after $10^6$ mini-batches

entropy. Although 17-step BPTT and UORO outperformed KeRNL, we expect speed-optimized versions of KeRNL to be much faster (wall clock speed) in direct comparisons.

To test how computation time for truncated-BPTT and KeRNL compare in the online setting, we implemented a dummy RNN, where the required tensor operations were performed using a random vector for both the input data and the error signal (Table 6, both algorithms were implemented in Python for uniformity; Details in Appendix A). KeRNL is faster than truncated BPTT beyond very short truncation lengths.

| Algorithm | KeRNL | 1 Step BPTT | 3 Step BPTT | 10 Step BPTT | 20 Step BPTT |
|---|---|---|---|---|---|
| CPU Time | 14.1 | 4.23 | 7.22 | 17.8 | 30.9 |

**Table 6:** Average CPU time (in units of $10^{-5}$ sec.) per time step for truncated BPTT and KeRNL.

# 6 CONCLUSIONS, DISCUSSION & FUTURE WORK

In this paper we show that KeRNL, a reduced-rank and forward-running approximation to back-propagation in RNNs, is able to perform roughly comparably to BPTT on a range of hard RNN tasks with long time-dependencies. One may view KeRNL as imposing a strong prior on the way in which neural activity from the past should be assigned credit for current performance, through the choice of the temporal kernels $K$ in the eligibility trace, and the choice of the sensitivity weights $\beta$. This product of two rank-2 tensors in KeRNL (replacing the rank-4 sensitivity tensor for backprop-agation in RNNs), assumes that the strength of influence of a neuron on another at fixed time-delay can be summarized by a simple sensitivity weight matrix, $\beta_{ki}$, [4], and a decay due to the time difference given by $K$. This strong simplifying assumption is augmented or mitigated by the ability to (meta)learn the parameters of the sensitivity weights and kernels in the eligibility trace, giving the rule simultaneously simplicity and flexibility. The form of the KeRNL ansatz or prior, if well-suited to learning problems in recurrent networks, serves as a regularizer on the types of solutions the network can find, and could even, for good choices of kernel $K$, provide better solutions than BPTT. We present limited evidence that KeRNL may combat the vanishing gradient problem with tanh units by imposing a prior of long time-dependencies through the eligibility. Finally, we show that KeRNL can be implemented online, where it has a shorter computation cycle than BPTT.

KeRNL is a step toward biologically plausible learning. It eschews the segmented two phase back-propagation algorithm for a computation that is largely feedforward. It does not require the segmentation and storage of all past states, instead using an integrated activity or eligibility trace, and it gives rise to a naturally asymmetric structure that is more similar to the brain.

While we show empirically that KeRNL performs hill-climbing, there is no guarantee that the gradients computed by KeRNL are unbiased. In the future, we hope to show empirically that KeRNL is able to perform well on more realistic tasks, and obtain some analytical guarantees on the performance of KeRNL. We hope the present contribution inspires more work on training RNNs with shorter, more plausible feedback paths. More generally, we hope that the present work shows how, with the use of reduced-rank tensor products and eligibility traces, to construct entire nested families of relaxed approximations to gradient learning in RNNs.

---

[4]This is not an entirely unreasonable assumption since we assume some static set of weights $W_{ki}^*$ can produce the desirable time-varying trajectory on the task.

ACKNOWLEDGMENTS

This work is supported in part by the HHMI through the Faculty Scholars' program, the Simons Foundation through the Simons Collaboration on the Global Brain, and CIFAR through the Senior Fellows program.

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

## A  IMPLEMENTATION DETAILS

For the adding problem and pixel-by-pixel MNIST, we tested performance by varying $\eta$ and $gc$ over several orders of magnitude: $\eta = \{1e-03, 1e-04, 1e-05, 1e-06, 1e-07\}$, $gc = \{1, 10, 100\}$, using both Adam (Kingma & Ba (2014)) and RMSProp (Tieleman & Hinton (2012)). We then varied $\eta_m = \{1e-03, 1e-04, 1e-05, 1e-06, 1e-07\}$ on KeRNL. We found that KeRNL was relatively robust across $\eta_m$. For all sequence lengths, we used the hyperparameters that performed best on the task with sequence length 400. Besides the recurrent weights of the IRNN, all other weight matrices were initialized using Xavier initialization (Glorot & Bengio (2010)). We initialized $\beta$ with Xavier initialization for the tanh RNN, and to the identity for the IRNN. This choice was motivated by the initial sensitivity of the IRNN ($\frac{\partial h_i^t}{\partial h_j^{t-\tau}} = \delta_{ij}$). For the kernels, we initialized $\gamma_j = (\frac{1}{\text{sequence length}})^{n_j}$ where the $n_j \sim U([0,2])$ are sampled uniformly and independently. Finally we used the alternative cost function described in footnote 2. We trained on both of these tasks using the Python numpy (Walt et al. (2011)) package.

For the dummy RNN, we used the Python numpy package (Walt et al. (2011)) to perform matrix algebra on a RNN with 100 hidden nodes, 100 input nodes and a tanh nonlinearity. We called "matmul" for matrix multiplication and "einsum" for other tensor operations. We used the "tanh" and "cosh" functions to compute the nonlinearity and its derivatives.

## B  KERNL FOR LSTM NETWORKS

In this section we describe how to implement KeRNL on an LSTM Hochreiter & Schmidhuber (1997) in more detail. The dynamics of the LSTM (without peepholes) are as follows

$$
\begin{aligned}
\mathbf{i}^t &= \sigma(W^{ii}\mathbf{x}^t + W^{ih}\mathbf{h}^{t-1} + b^i) \\
\mathbf{f}^t &= \sigma(W^{fi}\mathbf{x}^t + W^{fh}\mathbf{h}^{t-1} + b^f) \\
\mathbf{g}^t &= \tanh(W^{gi}\mathbf{x}^t + W^{gh}\mathbf{h}^{t-1} + b^g) \\
\mathbf{o}^t &= \sigma(W^{oi}\mathbf{x}^t + W^{oh}\mathbf{h}^{t-1} + b^o) \\
\mathbf{c}^t &= \mathbf{f}^t\mathbf{c}^{t-1} + \mathbf{g}^t\mathbf{i}^t \\
\mathbf{h}^t &= \mathbf{o}^t\tanh(\mathbf{c}^t)
\end{aligned}
\tag{12}
$$

where $\mathbf{h}^t$ is the hidden state, $\mathbf{c}^t$ is the cell state and $\mathbf{i}^t, \mathbf{f}^t, \mathbf{g}^t, \mathbf{o}^t$ are the input, forget, cell and output gates respectively. In order to implement KeRNL we consider the total hidden state $\mathbf{H} = \{\mathbf{h}, \mathbf{c}\}$ to be a concatenation of the hidden and cell states. This a suitable choice, as the next total hidden state can be fully determined by the current total hidden state and the parameters of the network. We let the first $n$ indices of $\mathbf{H}$ be the hidden state and the next $n$ be the cell state. Derivatives with respect to the cost function are given by

$$
\frac{dC}{d\theta} = \sum_{i,j} \frac{\partial C}{\partial H_i^t} \frac{\partial H_i^t}{\partial H_j^{t-\tau}} \frac{\partial H_j^{t-\tau}}{\partial \theta}
\tag{13}
$$

where $\theta$ stands in for the twelve trainable weights and biases. Our sensitivity Ansatz is $\frac{\partial H_i^t}{\partial H_j^{t-\tau}} = \beta_{ij}\exp(-\gamma_j\tau)$. The input terms are now partial derivatives of this total hidden state with respect to the input parameters. As an example

$$
\frac{\partial H_j^t}{\partial W_{jk}^{fi}} = \begin{cases} \frac{\partial h_j^t}{\partial W_{jk}^{fi}} & \text{if } j \le n \\ \frac{\partial c_{j-n}^t}{\partial W_{jk}^{fi}} & \text{if } j > n \end{cases} = \begin{cases} 0 & \text{if } j \le n \\ \frac{\partial c_{j-n}^t}{\partial W_{jk}^{fi}} & \text{if } j > n \end{cases}
\tag{14}
$$

where $\frac{\partial h_j^t}{\partial W^{fi}} = 0$ since the hidden state only depends on these parameters through the cell state. As earlier, we train our input weights and kernels by tracking the effect of applying perturbations during the forward pass. Our sensitivity weights $\beta$ are a 2 x 2 array of matrices linking the cell and hidden states of the past to the current cell and hidden states. Since readout occurs from the hidden state, $\frac{\partial C}{\partial c_i^t} = 0$, and we only need to consider $\frac{\partial h_i^t}{\partial c_j^{t-\tau}} = \beta_{ij}^{hc}\exp(-\gamma_j^c\tau)$ and $\frac{\partial h_i^t}{\partial h_j^{t-\tau}} = \beta_{ij}^{hh}\exp(-\gamma_j^h\tau)$. The sensitivity weights $\beta^{hc}$, $\beta^{hh}$ and time scales $\gamma^c$, $\gamma^h$ can be learned as in the case of the simple recurrent network by applying perturbations $\xi^{h,t}, \xi^{c,t}$ to hidden and cell state and minimizing the cost function: $C_{\beta,\gamma} = \sum_i(\tilde{h}_i^t - h_i^t - \sum_{\tau,j}\beta_{ij}^{hc}\exp(-\gamma_j^c\tau)\xi_j^{c,t-\tau} - \sum_{\tau,j}\beta_{ij}^{hh}\exp(-\gamma_j^h\tau)\xi_j^{h,t-\tau})^2$. Our example gradient with respect to the input weights of the input gate is given by

$$
\frac{dC}{dW_{jk}^{fi}} = \sum_i \frac{dC}{dh_i^t}\beta_{ij}^{hc}\sum_{\tau=0}\exp(-\gamma_j^c\tau)c_j^{t-\tau-1}\sigma'(\text{net}_j^{t-\tau})x_k^{t-\tau}
\tag{15}
$$

where net$_j^t$ represents the presynaptic input to $f_j^t$. The other gradients can be calculated in an analogous manner.

## C  KERNL ON PENNTREEBANK

In the interest of full disclosure, we note that KeRNL did not perform well on next word prediction on the Pen-nTreebank dataset. We tested an LSTM network across a wide variety of learning rates and gradient clippings and were not able to achieve near state of the art performance using KeRNL.

