# OpenReview forum: "Kernel RNN Learning (KeRNL)"
_ICLR.cc/2019/Conference_

### Official Review · AnonReviewer2 · 2018-11-01
**An interesting idea of improving BPTT by kernel recurrent learning. Skip in backpropagation is proposed and illustrated.**

**Rating:** 6
**Confidence:** 4

**Review:**

The proposed kernel recurrent learning (KeRL) provides an alternative way to train recurrent neural network with backpropagation through time (BPTT) where the propagation of gradients can be skipped over different layers. The authors directly assume the sensitivity function between two layers with a distance of tau in a form of Eq. (7). The algorithm of BPTT is then approximated due to this assumption. The model parameters are changed to learn the network dynamics. The optimization problem turns out to estimate beta and gamma of the kernel function. The learned parameters are intuitive. There are a set of timescales to describe the memory of each neuron and a set of sensitivity weights to describe how strongly the neurons interact on average. The purpose of this study is to save the memory cost and to reduce the time complexity for online learning with comparable performance.

Pros:
1. KeRL only needs to compute a few tensor operations at each time step, so online KeRL learns faster than online BPTT for the case with a reasonably long truncation length.
2. Biologically plausible statements are addressed.
3. A prior is imposed for the temporal sensitivity kernel. The issue of gradient vanishing is mitigated.
4. Theoretical illustration for KeRL in Sections 3 and 4 is clear and interesting.

Cons:
1. The proposed method is an approximation to BPTT training. Suppose the system performance is constrained. Some guesses are made. The system performance can be further improved.
2. The experiment on time cost due to online learning is required so that the reduction of time complexity can be illustrated.
3. The format of tables 1 and 2 can be improved. Caption is required in Table 1. Overlarge size of Table 2 can be fixed.
4.  A number of assumptions in Sections 3 and 4 are assumed.  When addressing Section 3, some assumptions in Section 4 are used. The organization of Sections 3 and 4 can be improved.

---

> ### Author Response · Authors · 2018-11-26
> **Response to Reviewer 2**
>
> Thank you for summarizing the positives features of KeRL while offering helpful critiques.
>
> We may add that KeRL might also work in places that BPTT wouldn’t: Imagine a scenario where the network is tasked with processing large amounts of real-time data quickly, and there is a speed/accuracy tradeoff. An algorithm like KeRL, which sacrifices accuracy for speed, should outperform BPTT.
>
> With respect to the specific critiques, we have fixed the tables and added an experiment on the time complexity of KeRL vs. BPTT. Finally, we have extensively edited sections 3-4 so for clarity and to emphasize the more important parts of the derivation.

---

### Official Review · AnonReviewer1 · 2018-11-04
**Interesting evidence that extreme approximations to BPTT can work**

**Rating:** 7
**Confidence:** 4

**Review:**

This paper proposes a simple method for performing temporal credit assignment in RNN training.  While it seems somewhat naive and unlikely to work (in my opinion), the experimental results surprisingly show reasonable performance on several reasonably challenging artificial tasks.

The core of the approach is based on equation 7, which approximates the Jacobian between different hidden states at different time-steps as a single adaptively-learned matrix times a decay factor that depends on the time gap.  While this seems like a very severe approximation to make the authors speculate that some kind of feedback alignment-like mechanism might be at play.

The presentation needs work in several areas, and the experimental results require more explanation, but otherwise this seems like a solid paper.  I would probably increase my rating if the authors could address my issues satisfactorily.


See below for more detailed comments:

Abstract & Section 1:

Is "sensitivity tensor" or "credit assignment tensor" common term?  Because I've never heard them before.  Consider defining them before you discuss it, and using consistent jargon.  Later in Section 2 you seem to call this the "RTRL tensor" (whose meaning I can infer).

Section 2:

Gradient vanishing isn't so much a problem in itself, but a symptom that the sensitivity of the network's output to the action of some neuron in the past is very low. Ths gradient is just relaying this information, so I don't really see vanishing gradients as the problem to overcome, but rather low sensitivity on past activations.

Section 3:

Did you mean to write (W^out h^t + b^out) instead of (W^out h + b^out)^t ?

"[equation] represents the gradient of the cost with respect to the current hidden state".  The RHS of this equation makes no sense to me.  Not only does this not depend on the nonlinearity in any way, it doesn't include any consideration of future outputs on which the current h surely depends.

It would make the paper much more pleasant to read if you gave your derivation of the learning rule before you stated it in gory detail.  It feels almost completely arbitrary reading it first without any justification. This might be fine if it were compact and elegant, but it's not.

Consider using exp(x) instead of e^x since the symbol e already means something else in your notation.

Section 4:

Please define "temporal variation"


Section 5:

You should elaborate on the experimental setup you used.  Especially for the Addition and MNIST problems.  For example, what consistutes a "step" in figure 2?  Does KeRL take "one" step per time-step?  Or does "step" mean a complete gradient computation from running from t = 1 to t= T?  Is the BPTT truncated?  Are you counting one step of BPTT to be one complete forwards and backwards pass?

You should include some basic description of what an IRNN is.

When you say that for MNIST that KeRL "does not converge to as good of an optimum" this seems like unjustified inference.  You don't really know that it is converging to a minimum of the original objective at all.  It could be converging to the minimum of some other objective it is implicitly optimizing due to your approximations (if one even exists).  Or it could be simply cycling around and failing to converge.  The fact that the loss plateaus isn't direct evidence of convergence in any sense.  If you wanted to measure this more directly you could look at the (true) gradient magnitude.

"only requires a few tensor operations at each time step" -> this is also true of UORO

---

> ### Author Response · Authors · 2018-11-26
> **Response to Reviewer 1**
>
> Thank you for the detailed review comments and criticisms, which were extremely helpful in improving our paper.
>
> Understanding the Ansatz is indeed key to understanding KeRL, and indeed giving various implementational details at the beginning obscured where the rule comes from. We now begin by first more clearly stating the learning rule, which actually does have a very simple form (this simple form was previously obscured by poor notation; we have now simplified the notation and also explain why the rule is simple), and then immediately showing the Ansatz that leads from the gradient-descent chain rule computation to KeRL. We have extensively edited the presentation of the Ansatz for clarity, and moved details on how to train the feedback weights and inverse-timescales to a different section.
>
> As for the detailed suggestions below, we have corrected the noted typos, clarified or added definitions where suggested, and replaced non-standard terminology. We think that our paper should be substantially clearer and the central idea easier to follow after implementing these suggestions.
>
> See below for more detailed comments:
>
> 'Abstract & Section 1:  Is "sensitivity tensor" or "credit assignment tensor" common term?  Because I've never heard them before.  Consider defining them before you discuss it, and using consistent jargon.  Later in Section 2 you seem to call this the "RTRL tensor" (whose meaning I can infer).'
>
> Replaced “credit assignment tensor” and “RTRL tensor” (both used interchangeably before) with the more clearly defined single term, “sensitivity tensor”, which has some precedence in the literature (please see paper text).
>
> 'Section 2: Gradient vanishing isn't so much a problem in itself, but a symptom that the sensitivity of the network's output to the action of some neuron in the past is very low. Ths gradient is just relaying this information, so I don't really see vanishing gradients as the problem to overcome, but rather low sensitivity on past activations.'
>
>  Removed this comment about gradient vanishing.
>
>  'Section 3: Did you mean to write (W^out h^t + b^out) instead of (W^out h + b^out)^t ? '
>
> Corrected typo.
>
> ' "[equation] represents the gradient of the cost with respect to the current hidden state".  The RHS of this equation makes no sense to me.  Not only does this not depend on the nonlinearity in any way, it doesn't include any consideration of future outputs on which the current h surely depends.'
>
>   Corrected.
>
> 'It would make the paper much more pleasant to read if you gave your derivation of the learning rule before you stated it in gory detail.  It feels almost completely arbitrary reading it first without any justification. This might be fine if it were compact and elegant, but it's not.'
>
> Please see response above, and fully edited presentation of rule in paper text.
>
> '  Consider using exp(x) instead of e^x since the symbol e already means something else in your notation'
>
> Done
>
> 'Section 4: Please define "temporal variation" '
>
> Clarified.
>
> 'Section 5: You should elaborate on the experimental setup you used.  Especially for the Addition and MNIST problems.  For example, what consistutes a "step" in figure 2?  Does KeRL take "one" step per time-step?  Or does "step" mean a complete gradient computation from running from t = 1 to t= T?  Is the BPTT truncated?  Are you counting one step of BPTT to be one complete forwards and backwards pass? '
>
> Clarified and elaborated. Also, replaced “training steps” with  “number of minibatches."
>
> ' You should include some basic description of what an IRNN is.'
>
>  Done
>
> 'When you say that for MNIST that KeRL "does not converge to as good of an optimum" this seems like unjustified inference.  You don't really know that it is converging to a minimum of the original objective at all.  It could be converging to the minimum of some other objective it is implicitly optimizing due to your approximations (if one even exists).  Or it could be simply cycling around and failing to converge.  The fact that the loss plateaus isn't direct evidence of convergence in any sense.  If you wanted to measure this more directly you could look at the (true) gradient magnitude. '
>
> Replaced with “does not reach as high an asymptotic performance”
>
>  ' "only requires a few tensor operations at each time step" -> this is also true of UORO'
>
> Clarified.

---

### Official Review · AnonReviewer3 · 2018-11-05
**limited empirical evidence**

**Rating:** 5
**Confidence:** 1

**Review:**

The paper proposes an alternative to backprop through time for
training RNN models.

The paper is reasonably well written, but somewhat dense and hard
to follow.  The contribution seems novel.

The main issue is the empirical evaluation.  All of the tasks
(masked addition, pixel-by-pixel MNIST, and the AnBn problem)
are artificial.

In addition, the results on some of the tasks are mixed if not
in favor of BPTT.  I am not convinced that these results are enough
to showcase the practical advantages of KeRL.

I am willing to increase my score, if the authors address this
issue.

Detailed comments:

- The authors mention that BPTT is not biologically plausible.  Although
  reasonable, I don't get why this would be an argument against it.

---

> ### Author Response · Authors · 2018-11-26
> **Response to Reviewer 3**
>
> "The paper is reasonably well written, but somewhat dense and hard to follow."
>
> Thank you for the valuable comments. We have extensively edited the exposition to both better motivate the rule and make its derivation clearer and easier to read.
>
> "The contribution seems novel. The main issue is the empirical evaluation.  All of the tasks (masked addition, pixel-by-pixel MNIST, and the AnBn problem) are artificial. In addition, the results on some of the tasks are mixed if not in favor of BPTT. "
>
> Since KeRL performs stochastic gradient descent (SGD) with an approximate gradient, we do not expect it to outperform untruncated BPTT, which performs SGD with an exact gradient. Rather, the way to think about KeRL results is like the results on feedback alignment: relaxing the symmetric return weights in feedback alignment allows for learning by an approximation to SGD. Recurrent learning is notoriously harder than learning in feedforward networks;  it's notable that KeRL, which makes a strong approximation to the sensitivity, is able to perform almost as well across a variety of difficult tasks. Furthermore, KeRL holds the advantage over BPTT that the computation of the gradient does not scale with the length of the graph.
>
> "I am not convinced that these results are enough to showcase the practical advantages of KeRL.  I am willing to increase my score, if the authors address this issue."
>
> We disagree that that the empirical evidence for KeRL is lacking. The adding problem was one of Schmidhuber’s “pathological” tasks that was used to demonstrate the utility of the LSTM. Using KeRL, we were able to solve sequence length 400 with a regular RNN and a squashing tanh nonlinearity. In fact, KeRL outperformed BPTT with the tanh nonlinearity, as the long timescales in the Ansatz were able to regularize the network towards a solution with longer sensitivity timescales. The pixel-by-pixel MNIST test, an even more challenging long term memory task as it involves remembering over nearly 1000 timesteps, was also solved on an RNN with KeRL.
>
>  "Detailed comments:  - The authors mention that BPTT is not biologically plausible.  Although reasonable, I don't get why this would be an argument against it."
>
> We are concerned with biological plausibility for three reasons. 1) From a neuroscience perspective we want to understand how the brain's recurrent networks learn tasks without the machinery to do BPTT; 2) Given that biological brains still outperform AI/deep networks on a wide range of problems, it is important to understand how brains solve these problems to build better AI, not just as a biological curiosity. 3) Designing and evaluating biologically realistic learning rules can be seen as an application of machine learning to neuroscience, one of the relevant topics of ICLR.

---

### Meta-Review · Area_Chair1 · 2018-12-14
**accept**

**Confidence:** 4
**Recommendation:** Accept (Poster)

**Metareview:**

this submission follows on a line of work on online learning of a recurrent net, which is an important problem both in theory and in practice. it would have been better to see even more realistic experiments, but already with the set of experiments the authors have conducted the merit of the proposed approach shines.